# Epidemiology and Toxicology of Ciguatera Poisoning in the Colombian Caribbean

**DOI:** 10.3390/md18100504

**Published:** 2020-10-01

**Authors:** Roberto Navarro Quiroz, Juan Carlos Herrera-Usuga, Laura Maria Osorio-Ospina, Katia Margarita Garcia-Pertuz, Elkin Navarro Quiroz

**Affiliations:** 1CMCC-Centro de Matemática, Computação e Cognição, Laboratório do Biología Computacional e Bioinformática–LBCB, Universidade Federal do ABC, Sao Paulo 01023, Brazil; roberto.navarro@ufabc.edu.br; 2School of Medicine, Universidad de Cartagena, Cartagena 130001, Colombia; jherrerau@unicartagena.edu.co (J.C.H.-U.); lauu.the@gmail.com (L.M.O.-O.); 3School of Medicine, Universidad Remington, Medellin 050001, Colombia; 4School of Medicine, Universidad del Norte, Barranquilla 080001, Colombia; kgpertuz@gmail.com; 5Faculty of Basic and Biomedical Sciences, Universidad Simón Bolívar, Barranquilla 080001, Colombia; 6Centro de investigación e innovación en Biomoléculas, Clinica de la Costa, Barranquilla 080001, Colombia; 7School of Medicine, Fundación Universitaria San Martín, Puerto Colombia 081007, Colombia; 8Centro de investigación e innovación en Biomoléculas, Care4You, Barranquilla 080001, Colombia

**Keywords:** ciguatoxins, dinoflagellates, intoxication, mannitol

## Abstract

Ciguatera is a food intoxication caused by the consumption of primarily coral fish; these species exist in large numbers in the seas that surround the Colombian territory. The underreported diagnosis of this clinical entity has been widely highlighted due to multiple factors, such as, among others, ignorance by the primary care practitioner consulted for this condition as well as clinical similarity to secondary gastroenteric symptoms and common food poisonings of bacterial, parasitic or viral etiology. Eventually, it was found that people affected by ciguatoxins had trips to coastal areas hours before the onset of symptoms. Thanks to multiple studies over the years, it has been possible to identify the relation between toxigenic dinoflagellates and seagrasses, as well as its incorporation into the food chain, starting by fish primarily inhabiting reef ecosystems and culminating in the intake of these by humans. Identifying the epidemiological link, its cardinal symptoms and affected systems, such as gastrointestinal, the peripheral nervous system and, fortunately with a low frequency, the cardiovascular system, leads to a purely clinical diagnostic impression without necessitating further complementary studies; in addition, what would also help fight ciguatera poisoning is performing an adequate treatment of the symptoms right from the start, without underestimating or overlooking any associated complications.

## 1. Introduction

Ciguatera fish poisoning (CFP) is an intoxication due to eating contaminated fish with ciguatoxins. Ciguatoxins are lipid-soluble polyether compounds consisting of 13 to 14 heterocyclic rings whose synthesis is associated with various species of dinoflagellates [1]. The ladder-like polyethers are produced by polyketide synthases (PKS), and the synthesis of the ladder-like polyethers by PKS starts with acetyl CoA, which is incorporated into long polyethers through a series of sequential condensations with malonyl CoA that are performed by KS domains of the PKS [2]. The main genera associated with the synthesis of ciguatoxins is *Gambierdiscus* [3]. Ciguatoxins were first isolated in 1967, describing an empirical formula of C_35_H_65_NO_8_ by means of Ultra-Micro methods [4,5]. The structure that is frequently described in the Caribbean Sea is the Caribbean ciguatoxin (CTX-1) [6].

On the other hand, CFPs have a great impact on the tourist and gastronomic sector of the Colombian Atlantic and Pacific coast, describing it in the Colombian Caribbean as the main cause of the ciguatera consumption of barracuda meat (*Sphyraena barracuda*) and moray eel (*Gymnothorax moringa*) [7].

Regarding the first reported case of CFP in Colombia, caused by ingesting the meat of a *Seriola zonata* (Mitchill, 1815), highlights the probability of existence of dinoflagellates that synthesize ciguatoxins in the vicinity of the Departments of Bolívar and Sucre in the Colombian Caribbean, although the species of dinoflagellate that caused the intoxication was not identified [8,9].

There is little information on CFP in Colombia and the registration of its dynamics is difficult, given that Colombia is not required to register for the National Public Health Surveillance System—SIVIGILA. For this reason, this review aims to make a survey of the ciguatera register in the Colombian Caribbean and the development of cases recorded when it was possible to find this description.

### 1.1. Epidemiology in Colombia

CFP is prevalent throughout the world; it is estimated that there are more than 50,000 cases in the world (Table 1). In the Colombian territory, CFP is not part of the diseases of mandatory notification for the Colombian government, which makes it important for Colombia to generate knowledge about the distribution of species of dinoflagellates that synthesize ciguatoxins with the aim of identifying hot spots, allowing medical personnel in these zones to be prepared to distinguish the signs and symptoms of CFP and allow to give the treatment most in accordance with CFP [3].

A study carried out by Mancera-Pineda in 2014, where they evaluated the presence of potentially toxic dinoflagellates associated with floating organic material in San Andrés Islas (drif), indicated that drift is an important substrate for dinoflagellates (*Prorocentrum emarginatum* and *Sinophysis microcephala*) and, given its floating nature, it represents perhaps the most important vector for the dispersion of these toxic agents on the island [10].

Regarding the records of ciguatera poisoning on the island of San Andrés, two outbreaks were recorded in 1997, which affected 16 tourists and 9 residents, respectively. On the island of San Andrés, regarding the presence of epiphytic toxic dinoflagellates in the seagrass meadows of the northern and eastern sectors of the island in coastal waters, the most abundant species was *Ostreopsis ovata*, which had a density of 23 cells/grams dry weight [11].

On the other hand, an ichthyotoxin poisoning event is described in the fishing community of the city of Tasajera, Department of Magdalena. Ciguatera poisoning was diagnosed in seven individuals whose ages ranged from 17 to 53 years (63.4% of the age range of fishermen in the sector), with symptoms of vomiting (100%); muscle pain in the lower extremities (71,4%); abdominal spasms (85.7%); diarrhea (100%); numbness and tingling in the face, hands and feet (85.7%); dizziness (100%); and a rash (14.2%); symptoms in most cases disappeared within 8 to 12 days. The main cause of the ciguatera event was the consumption of barracuda (*Sphyraena barracuda*) and brown meat (*Gymnothorax moringa*) [12].

Between 2010 and 2014, 101 cases of ciguatera in the department of San Andrés and Providencia were notified in the Public Health Surveillance System of the National Health Institute, with no associated mortalities. Fourteen isolated cases and 87 cases associated with 21 outbreaks were reported. On average, each outbreak affected 4.1 people and 20.2 cases were reported annually [13].

In 2017, Steven and collaborators reported that the highest CFP incidence rates in the Caribbean occur in Antigua and Barbuda (219 per 100,000 (CFP cases per 100,000)), and the 2003–2013 average projected division rate is 100 to 250 div. mo^−1^ for the *Gambierdiscus* sp. composite incidence rates; however, this averages zero to 0.02 per 100,000 (CFP cases per 100,000) in Colombia [14].

In the Colombian Caribbean, more than 200 cases of ciguatera have been reported since 1968, without deaths [15]. The symptoms of Colombian CFP patients different from those described in other parts of the world, where 31% presented itching throughout the body and 48% presented a metallic taste in the Mouth.

### 1.2. Distribution of Dinoflagellates Associated with Ciguatoxin in the Colombian Coasts

Regarding the richness and abundance of dinoflagellates associated with the production of ciguatoxins in Colombia, a previous study carried out in 2015, to determine the composition and abundance of dinoflagellates associated with seagrasses, collected 18 samples on Isla de Barú, and found ten diatom genera and three dinoflagellate genera, namely, *Prorocentrum*, *Ostreopsis* and *Gambierdiscus*, which include toxigenic species related to ciguatera and diarrheic shellfish poisoning (Figure 1). *Prorocentrum lima* was the most abundant dinoflagellate with average cell densities of 52 ± 48 cells/g substrate wet weight [3].

Three seagrasses, *Thalassia testudinum*, *Syringodium filiforme* and *Halodule wrightii*, were found on Isla de Barú, and *T. testudinum* was the most abundant and dominant. Regarding the cell densities of the epiphytic dinoflagellates, there was no statistically significant difference between the months of study (in April and September of 2015) on the Island of Barú, and the same behavior was shown in the study sites; furthermore, findings on Isla de Barú suggest that El Niño may modulate dinoflagellate populations, increasing its abundance compared to neutral periods, especially in species of the genus *Prorocentrum* [3].

In 2010, Rodriguez et al. studied the island of San Andres for the dinoflagellates associated with *Thalassia testudinum,* and found seven different dinoflagellates (*Ostreopsis ovata*, *Prorocentrum emarginatum*, *Prorocentrum lima*, *Prorocentrum hoffmannianum*, *Prorocentrum maculosum*, *Prorocentrum rathymun* and *Sinophysis microcephala*) with a cell density mean of 166 cells/grams dry weight, divided between eight sites (Rocky Cay, Mar Azul, Bahía Honda, Harbor, Cotton Cay, Isleño, Toninos and Acurio) [11].

## 2. Pathophysiology

*Gambierdiscus toxicus* is the dinoflagellate most associated with the production of ciguatoxins that are the cause of systemic symptomatology [16]. These toxins are of a lipid nature, forming rings joined by ether bonds that gives them firmness in its structure. Three types of ciguatoxins have been identified, namely, CTX1B, 54-deoxyCTX1B and 52-epi-54-deoxyCTX1B, which accumulate more frequently in the muscle, liver, kidneys and spleen of bioaccumulated fish. These toxins present intrinsic resistance to cold, heat and exposure to acidic and basic media; therefore, cooking contaminated fish is not a guarantee for the prevention of toxic infection [17].

The intoxication is the result of eating herbivorous fish that have fed on the dinoflagellates of the genus *Gambierdiscus* that synthesize gambiertoxins. The biomagnification process of ciguatoxins in fish that have been fed with *Gambierdiscus* sp. have been previously described, establishing an association between the size of the herbivorous fish that habitually feed on this type of dinoflagellates and the concentration of the ciguatoxins, which is species specific [18].

Several factors influence the risk of suffering CFP, being more frequent when consuming fish of a large size or high longevity due to the fact that it is more likely that they have a greater accumulation of ciguatoxins given the associations made with these two factors previously [19]. It should be noted that the bioaccumulated fish have a normal appearance, texture, smell and taste [20].

As for the toxicokinetics of the ciguatoxins, it has been described that ciguatoxins are voltage-gate selective; the intestinal absorption of ciguatoxins is rapid, and it is quickly localized in soft tissues such as skeletal muscle, the heart and the nervous system. Ciguatoxins are transported together with plasma proteins, mainly seroalbumin, and can be found in toxic levels in body fluids such as breast milk and seminal fluid, in addition to their characteristic of liposolubility, making them able to cross the placental barrier and can cause fetal conditions and abortions during the acute phase of the poisoning. After a liver biotransformation, finally the toxin is excreted by bile in the feces [21].

Toxicodynamics consists of an induction to the opening of the voltage-gated sodium channels dependent on ciguatoxins at the level of skeletal muscle, heart cells and mainly peripheral nerves. Once the sodium channels open, an intracellular flow of sodium is generated that causes water to enter the cell and produce edema generated at the level of the white tissues by the toxin mentioned previously. In search of intracellular homeostasis, compensatory mechanisms begin the expulsion of excess sodium internalized in the cell, exchanging it for extracellular calcium to such an extent that the intracellular calcium levels can become so high that they generate increased tissue contraction force muscle [22].

### 2.1. Clinical Manifestations

The general signs and symptoms of CFP are as follows: headache, diarrhea, vomiting, osteotendinous hyporeflexia, hives, conjunctivitis, pain and decreased visual acuity. These clinical presentations may vary in the acute, chronic and relapsing phases of the illness [23].

According to the article of Palafox and Buenconsejo-Lum in 2001 [23], the clinical manifestations can be classified into three groups, which are (1) acute (within the first two weeks of exposure of ciguatoxin); (2) chronic (persisting beyond the two weeks of the initial intoxication); and (3) relapsing phases of the illness (occur years after the initial intoxication) [23].

Looking at the first symptoms detected, they frequently appear 4–6 h after toxin consumption, but can occur within minutes with large toxin ingestions or can manifest in 24 h with smaller doses of the toxin [24].

CFP presents a great variety of symptoms and symptomatological patterns, depending on the region where the contaminated meat has been ingested. It is said that the Caribbean areas have primarily gastrointestinal symptoms of the neurological type, whereas the opposite happens in the Pacific [25]; these differences in symptomatology in different geographic regions are probably related to structural differences between the ciguatoxins of the Pacific, the Caribbean and India. Symptoms usually begin minutes after ingestion; however, cases of late onset, e.g., after more than 36 h, have been described. For unknown reasons it has been observed that people who ingest fish contaminated with ciguatoxins do not present any condition [26]. More than 175 symptoms related to toxic infection have been described, and the following will emphasize the most common ones [27].

Gastrointestinal tract: It is perhaps the most affected and associated with earlier onset; it is characterized by diarrhea, nausea, vomiting, metallic taste, abdominal pain and pain with defecation. The gastroenteritis of infectious origin appears easily; however, they have very nonspecific symptoms that do not usually orient the ciguatera disease with certainty, and can take a long time to disappear, usually between 1 and 2 days after ingestion [28].

Neurological manifestations: Taking into account the pathophysiology of the toxin when producing sodium-channel opening, whereby large intracellular water flow causes edema at the level of the Schwann cells and the axons generate a decrease in the speed of the nerve conduction, and an increase of the refractory phase of the potential of the action, the patients present paresthesia in their limbs that could start from 30 min or days after the consumption of the contaminated fish, being able to persist for months or years. In addition, cases of odontalgias have been reported, so too ataxia, vertigo, allodynia and headache after consumption of ciguatoxins [29,30,31].

Cardiovascular manifestations: These are the least frequent but the most feared, usually starting 2 and 3 days after consumption [32]. The toxin has a direct effect on the papillary muscles and on the atrium, therefore predisposing the patient to presenting cardiac arrhythmias, extrasystoles and heart failure that can be potentially lethal. Orthostatic hypotension due to stimulation of the parasympathetic system and respiratory arrest due to phrenic nerve block has also been described [33].

Other manifestations: The most frequent skin conditions of toxic infection are a rash associated with pruritus and are related to acne exacerbations [14].

The main route of exposure of the toxin is orally; there are other ways of transmission, such as maternal–fetal, for which cases of abortions and preterm birth have been reported. In addition to the oral route is the sexual route, by the presence of the toxin in the seminal fluid, causing pain during ejaculation, vaginal burning and pelvic pain; these symptoms may persist for 2 to 3 weeks [34].

### 2.2. Diagnosis

The diagnosis is clearly clinical, looking at the recent history of fish consumption, more so if they are species of reefs, and at the subsequent onset of a gastroenteric condition associated with neuropathic symptoms, mainly at the level of the peripheral nervous system. It should always be remembered that the consumption of fish in non-coastal areas does not exclude the diagnosis [35].

The results of general paraclinics, such as a hemogram, arterial gases, and renal and hepatic function, generally yield results in the normal range. Nonspecific alterations in heart rhythms documented in electrocardiogram traces should generate suspicion about the cardiac compromise generated by the ciguatoxins [36].

So far there is no specific paraclinical method for accurate diagnosis of ciguatera in humans. A practical product is commercialized to detect toxins in the meat of the fish to be consumed, called ciguaCheck [36,37]. In addition to this, in 2018, a highly sensitive fluorescent sandwich ELISA, which can detect, differentiate and quantify the four major CTX congeners (CTX1B, CTX3C, 51-hydroxyCTX3C, and 54-deoxyCTX1B [38]) with a detection limit of less than 1 pg/mL, was successfully developed by Tsumuraya and Hirama [39].

It is worth mentioning that the most traditional biological method that is available for use in most care centers, having considerable reliability without being highly specific, is not considered a routine diagnostic method due to its high cost; so, it is not cost effective. From a mouse bioassay, where the rodent was fed contaminated fish meat, it was learnt that one of its main disadvantages is the long observation time of the mice that consumed the contaminated fish meat [40,41].

### 2.3. Differential Diagnoses

Like ciguatera, there is a wide range of toxic infections related to consumption of seafood that are the same or mostly unknown and whose clinical manifestations can be very similar [42].

Tetradotoxism caused by tetradotoxin, which is acquired by consuming puffer fish, is the most fatal marine toxin causing systemic block of the sodium channels, generating symptoms from approximately 20 min up to 3 h after ingestions, rapidly evolving into nausea, vomiting, headache, paresthesia, dysarthria, ataxia, quadriplegia, respiratory failure, coma and death. There is no evidence of an antidote, and treatment consists of limiting or minimizing absorption and treating complications that threaten the patient’s life [43].

The scombroidosis caused by the consumption of fish that have not had a good refrigeration begins a process of decomposition with growth of bacterial colonies activating the enzyme histidine decarboxylase using as a substrate histidine, abundant amino acid in the muscle tissue of reef fish [44], with the end product of the enzymatic activity being a large amount of histamine generated, which is absorbed and distributed at the systemic level of the person affected, generating symptoms such as flushes, rashes, pruritus, nausea, vomiting, diarrhea, headache, conjunctival, cough, bronchospasm, tachycardia and anaphylactic shock [45].

## 3. Treatment

Treatment consists of adrenaline in cases of anaphylactic shock, corticosteroids and antihistamines [46].

Additionally, intravenous mannitol remains the primary treatment consideration for CFP. Mannitol therapy has been recommended for the goal of reducing symptoms (especially neurologic) during the acute stage of the illness; it can be used in patients with significant morbidity due to poisoning by ciguatoxins and it is recommended to administer 1.0 g/kg body weight over a 30–45 min period [1,47,48].

Food poisoning of bacterial, viral or parasitic etiology can be triggered by consumption of contaminated fish during the process of handling these foods, often associated with dysentery and fever, and are not related to neurological symptoms. except for botulism caused by intake of *Clostridium botulinum* toxins [1].

The clinical management of ciguatera poisoning is symptomatic and supportive; it should be noted that the topic should be known and that it exists so as not to underreport such poisoning. In the following section, a division between acute management and chronic management will be performed [1].

### 3.1. Acute Management

Patients with symptoms such as a skin rash, pruritus and acute gastroenteritis: Medical management is with isotonic intravenous fluids, such as 0.9% saline or Ringer’s lactate, depending on the state of dehydration of the patient; in addition, an antiemetic and an antihistamine can be used [49].

There are case reports in which patients have severe poisoning, defined as alterations in the state of consciousness, presence of cardiac arrhythmias and/or hypotension; in these cases one must start by ensuring the airway is open if necessary, starting mannitol early at a dose of 1gr/kg of weight to pass in 30 min to 1 h and the dose can be repeated. As such, its mechanism of action is not known exactly but it has been speculated that due to its osmotic diuretic effect, it sweeps the ciguatoxins and decreases the axonal edema that causes competitive inhibition with the sodium channels. To begin the said diuretic, one should not wait more than 72 h [49].

In cases of hypotension, supportive medical management with dopamine and anti-shock therapy with volume expanders can be initiated; in case of bradycardia, atropine at a dose of 0.5–2 mg intravenously is useful [50].

### 3.2. Chronic Management

As for chronic management, it should be noted that all treatments are aimed at neuropathic therapies, since the predominant symptoms are from the peripheral nervous system; for example, paresthesia, dysesthesia and vertigo; multiple drugs have been studied for these symptoms, and currently they are used with little clinical evidence (e.g., pregabalin, gabapentin, calcium channel inhibitors, such as nifedipine, and amitriptyline as a sodium channel membrane stabilizer). It must be said that the management of chronic symptoms could be long term and, in some circumstances, it becomes a challenge for the clinician [51].

## 4. Conclusions

Ciguatera poisoning is a disease with great impact on public health, especially in the Colombian coast. The most frequently given advice is not to consume fish weighing more than 2 kg, avoid eating fish such as barracuda and not eating fish parts such as the viscera, brain and gonads, which is where ciguatoxins is mostly accumulated. An invitation is sent to the territorial entities to carry out control measures for the consumption of certain fish; in addition, because this is a medical and environmental alert, the symptoms must be known and an adequate diagnosis of this poisoning must be given, so that CFP does not drastically disrupt our daily life.

## Figures and Tables

**Figure 1 marinedrugs-18-00504-f001:**
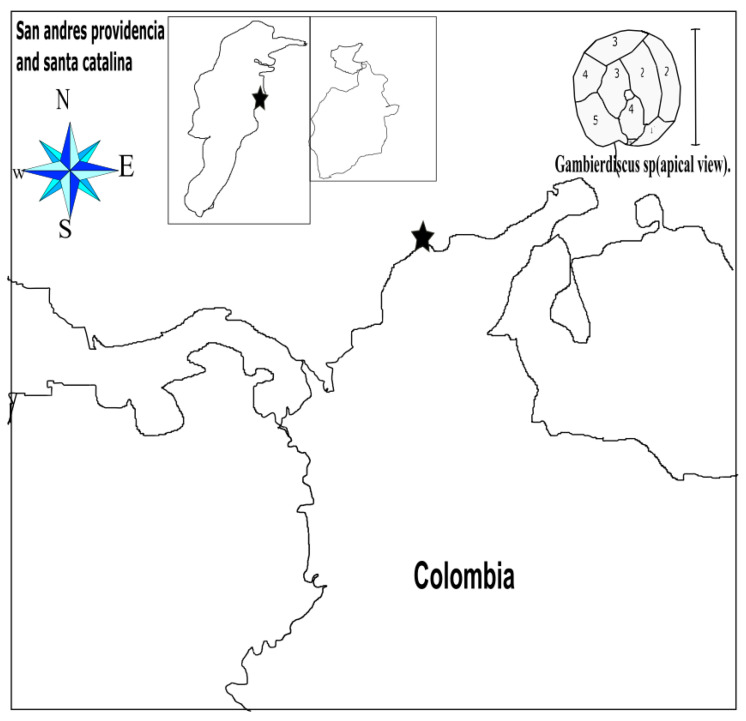
Colombian Atlantic coast and, at the top, a dinoflagellate representative of the *Gambierdiscus* sp. (for each of the 6 plates that are distinguished in the apical view, the bar represents 200 micrometers). The stars represent places where ciguatoxin fish poisoning (CFP) cases have been reported.

**Table 1 marinedrugs-18-00504-t001:** Number of outbreaks and cases of ciguatera reported between 1968 and 2015.

Year	Number of Shoots	Number of Cases Associated with Outbreaks	Number of Isolated Cases	Total
**1968**	1	28	0	28
**1984**	1	15	0	15
**1994**	1	7	0	7
**1997**	2	25	0	25
**2005**	1	7	0	7
**2007**	2	25	0	25
**2010**	5	28	0	28
**2011**	1	9	0	9
**2012**	3	12	2	14
**2013**	9	30	3	33
**2014**	3	8	9	17
**2015**	1	30	0	30
**Total**	30	224	14	238

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
