# Peer review of "Epidemiology and Toxicology of Ciguatera Poisoning in the Colombian Caribbean"

_marinedrugs, 2020, doi:10.3390/md18100504_

Round 1

Reviewer 1 Report

The review article reported by E. N. Quiroz et al. summarizes the epidemiology and toxicology of ciguatera food poisoning in the Colombian Caribbean.  The information reported in the article is of great interest to the researcher in the field of ciguatera seafood poisoning.  However, I am sorry to say that the review article presented by E. N. Quiroz et al. still does not meet the standards required for a publication in Marine Drugs.  In addition, the authors should clearly respond or refute each and every question posed to them when they resubmit, which is unclear.  I, therefore, advise the rejection of the manuscript as in the present form.  Again I pointed out several points that should be considered to improve the manuscript.

  • I advise the authors to get editing help from someone with full professional proficiency in English.
  • In the introduction section, the authors need to comment on the details of ciguatoxins. This includes chemical formulae of ciguatoxins.  This information is very useful for readers.  The following pieces of literature are for CTXs in the Pacific regions. The authors do not need to cite all of them, but I think they should cite some of them.

Murata, M.; Legrand, A.M.; Ishibashi, Y.; Yasumoto, T. Structures of Ciguatoxin and Its Congener. J. Am. Chem. Soc. 1989111, 8929–8931.

Murata, M.; Legrand, A.M.; Ishibashi, Y.; Fukui, M.; Yasumoto, T. Structures and Configurations of Ciguatoxin from the Moray Eel Gymnothorax javanicus and Its Likely Precursor from the Dinoflagellate Gambierdiscus toxicus. J. Am. Chem. Soc. 1990112, 4380–4386.

Satake, M.; Murata, M.; Yasumoto, T. The structure of CTX3C, a ciguatoxin congener isolated from cultured Gambierdiscus toxicus. Tetrahedron Lett. 199334, 1975–1978.

Satake, M.; Fukui, M.; Legrand, A.-M.; Cruchet, P.; Yasumoto, T. Isolation and structures of new ciguatoxin analogs, 2,3-dihydroxyCTX3C and 51-hydroxyCTX3C, accumulated in tropical reef fish. Tetrahedron Lett. 199839, 1197–1198.

Yasumoto, T.; Igarashi, T.; Legrand, A.-M.; Cruchet, P.; Chinain, M.; Fujita, T.; Naoki, H. Structural elucidation of ciguatoxin congeners by fast-atom bombardment tandem mass spectroscopy [5]. J. Am. Chem. Soc. 2000122, 4988–4989.

  • The authors use the nomenclature of ciguatoxins in Pacific regions, like P-CTX1, P-CTX2, and P-CTX3. I recommend using the nomenclature suggested by Prof. Yasumoto, who has first reported the chemical structure of ciguatoxin.   Instead of P-CTX, use CTX1B; instead of P-CTX2, use 52-epi-54-deoxyCTX1B; instead of P-CTX3, use 54-deoxyCTX1B. There are two reasons that I recommend using Yasumoto’s nomenclature.  One is Yasumoto discovered the first structure of CTX, we should follow his recommendations.   Another reason is although CTX1B was first discovered in the Pacific but also later discovered in the Atlantic.  So, the name Pacific CTX is not appropriate.   This is my opinion but please consider it.
  • The author mentioned CiguaCheck discovered by Hokama et al. However, the results of the method have been in a controversy (for example, please see Toxicon 46 (2005) 563-571.)  I recommend the authors include recently reported sandwich ELISA detection of CTXs by Tsumuraya and Hirama. (for example, Analytical Chemistry 90 (2018), 7318)
  • The author should include any comments on the difference between the CFP that occurred in Colombia and those that occurred in other areas.
  • The usages of the terms ciguatoxin (CTX) and ciguatera fish poisoning (CF)) are incorrect.  Please check the manuscript thoroughly and correct them prroperly.

Author Response

Sincerest thanks for your response and reviewers comments on our manuscript. Following your letter regarding the manuscript “ Epidemiology and toxicology of Ciguatera poisoning in the Colombian Caribbean

Reviewer 2 Report

The manuscript entitled “Epidemiology and toxicology of Ciguatera poisoning in the Colombian Caribbean” is focused to give the picture of Ciguatera poisoning in Colombia.  Ciguatera is currently one of the “hot-topic” in the field of marine toxins as it has become a global issue. The review is not well written and needs some language editing. Many aspects of the Ciguatera issue seem to be ignored, especially the chemistry of different CTX analogues (in fish) and their biosynthetic precursors (in microalgae). I strongly suggest a reorganization/language editing/ enrichment in contents (especially Introduction). Here follow just some specific comments:

I was suggesting to modify “Ciguatotoxin poisoning" into "Ciguatera fish poisoning"(referring to the syndrome due to human consumption of contaminated fish!!) but probably a misunderstanding occurred and the Authors “decided” to use  “Ciguatera fish poisoning (CFP)” to define the toxins, that actually it’s not what I suggested to do! Ciguatoxins are the compounds responsible for Ciguatera fish poisoning (and sometimes in the text the Authors use the misspelled term, ciguatotoxins).  So, all over the text the Authors should use “Ciguatoxins” when they refer to compounds, whereas they have to use “Ciguatera fish poisoning” when they refer to the human poisoning. I have tried to spot out the points in the text where it seems to me that the Authors are referring to the toxins, and I highlight it in yellow in the pdf file.

Line 16: The sentence: “Ciguatera is intoxication for ate contaminated fish with dinoflagellate’s ladder frame polyether  toxins, Ciguatera fish poisoning (CFP) are polyether ladder-formed toxins produced by several  species of diatom and dinoflagellate [1], the main genus associated with ciguatoxins produced is  Gambierdiscus, addition to Gambierdiscus in colombia have been reported other genus such as: Actinocyclus, Nitzschia, Rhabdonema, and Thalassionema” has to be rephrased.

Line 20: Please rephase the sentence “CFP were first isolated in 1967 from  Gymnothorax javanicus[3], in 1989 the Ciguatera was postulated that its accumulated in fish  throughout the food webs, as well as, it was elucidated the structure of CFP[4]. The structures more described is Caribbean ciguatoxin (C-CTX-1) and its molecular formula is C62H92O19 [5].”

Line 24:  turn “On the other hand, the epidemiological(affects up to 50,000 people each year throughout the  world) effects of ciguatera….” into “On the other hand, the epidemiological effects of ciguatera (CFP affects up to 50,000 people each year throughout the world)…”

Line 29:  turn “ichthyosarcotoxicosis” into “ichthyosarcotoxicosis (CFP)”

Line 29. It’s not clear what the Authors mean with the following sentence ”Regarding the first historical record of the reported case of ichthyosarcotoxicosis in Colombia, caused by ingesting the meat of a slender amberjack (Seriola zonata) does not leave on the presence of dinoflagellates with toxic properties in waters adjacent to the departments of Bolívar and Sucre in  the Caribbean Colombian”

 Line 33 : The Authors say a trolley, do they mean a fish trap?

Line 49: Turn “in San  Andrés Islas(drif),” into “in San Andrés Islas (drift).”

Line 65: The Authors stated  “In April 2007, in San Andrés, two ciguatera outbreaks occurred with 9 and 16 people involved  (residents and tourists respectively) after ingesting barracuda” whereas at line 52-53, they reported “Regarding the records of ciguatera poisoning on the island of San Andrés , two outbreaks were recorded in 1997, which affected 16 tourists and 9 residents”. Could the Authors check that it is not a duplicate?

Line 73: there is no need to acronyms again CFP

Line 85, 91: Prorocentrum, Ostreopsis, and Gambierdiscus, Thalassia testudinum, Syringodium filiforme and Halodule wrightii : Genus/species names have to be in italics (check this all over the text )

Line 91: Turn “Three species of seagrasses” into “Three seagrasses”

Line 93: The sentence “Although the cell densities of epiphytic dinoflagellates found on Isla de Barú in April and September of 2015 did not show statistically significant differences between sites or months of sampling 94 (p>0.05)[2].” seems incomplete.

Line 97: Please turn “they found seven species of dinoflagellates (Ostreopsis ovata, Prorocentrum  emarginatum, Prorocentrum lima, Prorocentrum hoffmannianum, Prorocentrum maculosum, Prorocentrum rathymun and Sinophysis microcephala )” into “they found seven different dinoflagellates (Ostreopsis ovata, Prorocentrum emarginatum, Prorocentrum lima, Prorocentrum hoffmannianum, Prorocentrum maculosum, Prorocentrum rathymun and Sinophysis microcephala )”.

Figure 1. In my opinion, the quality of the figure is not enough, and it should be improved. Furthermore, Providencia, Santa Catalina and Saint Andres that are three different islands are not distinguishable.

Line 110: The sentence “Identified three types of ciguatotoxins P-CTX1, P-CTX2, P-CTX3 which accumulate more frequently in muscle, liver, kidneys, and spleen of bioaccumulated fish” seems a nonsense sentence.

Line 110: Please turn “ciguatotoxins” into “ciguatoxins”

Line 114: Please rephrase the following sentence: “The intoxication is result of eat herbivorous fish that its main form food are dinoflagellates and their toxins are consumed and bioaccumulate by these seconds [16]”

Line 116: The sentence must be refined: “Several factors influence the risk of acquiring of being intoxicated, being more viable when consuming large or long-lived fish because they are the largest accumulation of toxins”

Line 138-141: The Authors stated: “manifestations are divided space time in:”. Rewording is encouraged.

Line 145-146: The sentence would need some refinements: “The disease given by ciguatotoxin presents a great variety of symptoms whose behavior may even depend on the region where it is ingested, it is said that the Caribbean areas have primarily  gastrointestinal symptoms on the neurological ones, the opposite happens in the Pacific”. The Authors should describe the reasons why the differences in symptomatology in different geographical regions exist, likely linked to the structural differences between Pacific, Caribbean and Indian ciguatoxins. Very few structural info is provided ..

Line 153-156 also this sentence would need some refinements: “Easily appearing a gastroenteritis of infectious origin, they are very nonspecific symptoms that usually do not orient the ciguatera disease with certainty, they can take a long time to disappear, usually between 1 and 2 days post-intake”

Line 165: Turn “Manifestation” into “manifestations”

Author Response

Sincerest thanks for your response and reviewers comments on our manuscript. Following your letter regarding the manuscript “ Epidemiology and toxicology of Ciguatera poisoning in the Colombian Caribbean

We sent this letter with information change step to step. Please see the attachment.

Round 2

Reviewer 1 Report

The authors fully address my concerns.   I, therefore, recommend the manuscript for publication on Marine Toxins.  However, the following minor points should be revised before publication.

-In line 221, "Tsuburaya" should be corrected to "Tsumuraya".   Please see reference 39.

-In line 132, CTX1 should be removed, because CTX-1 is CTX1B.  Also, 52-epi-54-deoxy should be corrected to 52-epi-54-deoxyCTX1B.

Author Response

Sincerest thanks for your response and reviewers comments on our manuscript. Following your letter regarding the manuscript “ Epidemiology and toxicology of Ciguatera poisoning in the Colombian Caribbean

Response to Reviewer Comments 1

Point 1:  -In line 221, "Tsuburaya" should be corrected to "Tsumuraya".   Please see reference 39..

Response 1: in line 220 the author's name was corrected  "Tsumuraya"

Point 2: -In line 132, CTX1 should be removed, because CTX-1 is CTX1B.  Also, 52-epi-54-deoxy should be corrected to 52-epi-54-deoxyCTX1B.

Response 2: In lines 132,  it was change CTX1 by CTX1B as well as 52-epi-54-deoxy by 52-epi-54-deoxyCTX1B

Reviewer 2 Report

The Manuscript  aims to give an overview on the Ciguatera Fish Poisonig (CFP) phenomenon in Colombia. CFP is foodborne illness due to consumption of large reef fish that can bioaccumulate potent neurotoxins that reach humans through the food web.  The Authors put many efforts into revising their original draft but,   despite all this, I strongly suggest improving the language quality!

I used text callout tool for reviewing and in the attached pdf you’ll find a number of specific comments that can be explicative of my opinion.

Author Response

Sincerest thanks for your response and reviewers comments on our manuscript. Following your letter regarding the manuscript “ Epidemiology and toxicology of Ciguatera poisoning in the Colombian Caribbean

Response to Reviewer Comments 2

Point 1: In line 29, it not clear, please check grammar of sense  ” identify the relation of the dinoflagellates in the assembly of said toxin”.

Response 1: In line 29, the sentence was written to be grammatically correct

Point 2: In line 41 and 46, change the comma for full stop 

Response 2: In line 41 and 46, comma was changed to full stop point

Point 3:  In line 47, the word Gambierdiscus was changed to italic

Response 3:  In line 47, the word Gambierdiscus was changed to italic

Point 4: all sentence seen not Grammarly correct   “On the other hand, the epidemiological effects of ciguatera (CFP affects up to 50,000 people each 51 year throughout the world)Ciguatoxins effects of ciguatera have a great impact on the tourist and 52 gastronomic sector in the Colombian Atlantic and Pacific coast so that they have been described in  the Colombian Caribbean that main cause of the ciguatera event consumption of barracuda meat  (Sphyraena barracuda) and moray eel (Gymnothorax moringa)”

Response 4:  the sentence was rewritten

Point 5: line 57 , this sentence is incomplete.

 Response 5: this change was made as recommended

Point 6: and for ease as signs and symptom it not clear”

Response 6: that phrase was rewritten

Point 7: CFP is prevalent throughout the world, it is estimated that there are more than 50,000 cases in 65 the world, in Colombian territory CFP is not part of the mandatory notifiable diseases, which makes  it important for Colombia to generate knowledge about of the distribution of dinoflagellate species  that synthesize ciguatoxins, given that that the classification of sea areas is a crucial aspect for the  application of appropriate protocols for treatment in areas where the greatest abundance and richness  of these dinoflagellates are found and make CFP more likely to occur, the dinoflagellates that  synthesize ciguatoxins are around 50 species, however, it is assumed that their distribution is  cosmopolitan, in their habitat they cover surfaces of reefs and macroalgae” 

Response 7: this was modified

Point 8: A study carried out by the National University of Colombia department of biology evaluated 73 the check 

 Response 8 : this change was made as recommended

Point 9: in line 78 , it was added respectively

Response 9: on line 78, it was added respectively, and the first letter of the next word was changed

Point 10:  in line 98, it is better explained what the values correspond to

Response 10: it is better explained what the values correspond to

Point 11: in line 104 and 105 , check question mark and CPF

Response 11: : this was modified

Point 12: in line 111 and 112,. The temperature hypothesis gains strength as one of the main modulators of dinoflagellate abundance observed in the Caribbean, especially regarding Prorocentrum species and some diatoms  such as Mastogloia corsicana and Actinocyclus normanii 

Response 12 : this was modified 

Point 13: modify figure 1 place the geographic location.

Response 13: this was modified 

Point 14: in line  145 and 151 , As for toxicokinetics of Ciguatoxins, many details remain to be clarified, however it is known  that its intestinal absorption is rapid, it is quickly located in white tissues such as skeletal muscle,  heart and nervous system, it is transported together with plasma proteins, mainly seroalbumin, it can  be Finding toxin levels in body fluids such as breast milk and seminal fluid, in addition to its  characteristic of liposolubility, crosses the placental barrier and can cause fetal conditions with  aborted during the acute phase of the poisoning , After a liver biotransformation finally, the toxin is  excreted by bile and feces [21].
”.

Response 14: this was modified 

Point 15: The sentence, Additionally in initial treatment with mannitol may be considered, although the mechanism is.

Response 15: this was modified

Point 16: The sentence ,Patient with symptoms with skin rash associated with pruritus plus acute gastroenteritis, his  medical management is with isotonic intravenous fluids such as 0.9% saline solution or ringer's  lactate according to the state of dehydration in which the patient is associated to it is added a antiemetic plus antihistamines for local symptoms[48].

Response 16:that phrase was rewritten

Point 17: in line 281 Please turn “comma” into “full stop”

Response 17: this was modified 

This manuscript is a resubmission of an earlier submission. The following is a list of the peer review reports and author responses from that submission.

Round 1

Reviewer 1 Report

The review article reported by E. N. Quiroz et al. summarizes the epidemiology and toxicology of ciguatera food poisoning in the Colombian Caribbean.  The information reported in the article is of great interest to the researcher in the field of ciguatera seafood poisoning.  However, I am sorry to say that the review article presented by E. N. Quiroz et al. does not meet the standards required for a publication in Marine Drug.  I therefore advise the rejection of the manuscript as in the present form.  The following points should be considered to improve the manuscript.

1) I advise the authors to get editing help from someone with full professional proficiency in English.

2) In the introduction section, the authors need to comment on the details of ciguatoxins. This includes the chemical formulae of ciguatoxins.  This information is very useful for readers.  The following pieces of literature are for CTXs in the Pacific regions. The authors do not need to cite all of them, but, I think they should cite some of them.

Murata, M.; Legrand, A.M.; Ishibashi, Y.; Yasumoto, T. Structures of Ciguatoxin and Its Congener. J. Am. Chem. Soc. 1989111, 8929–8931.

Murata, M.; Legrand, A.M.; Ishibashi, Y.; Fukui, M.; Yasumoto, T. Structures and Configurations of Ciguatoxin from the Moray Eel Gymnothorax javanicus and Its Likely Precursor from the Dinoflagellate Gambierdiscus toxicus. J. Am. Chem. Soc. 1990112, 4380–4386.

Satake, M.; Murata, M.; Yasumoto, T. The structure of CTX3C, a ciguatoxin congener isolated from cultured Gambierdiscus toxicus. Tetrahedron Lett. 199334, 1975–1978.

Satake, M.; Fukui, M.; Legrand, A.-M.; Cruchet, P.; Yasumoto, T. Isolation and structures of new ciguatoxin analogs, 2,3-dihydroxyCTX3C and 51-hydroxyCTX3C, accumulated in tropical reef fish. Tetrahedron Lett. 199839, 1197–1198.

Yasumoto, T.; Igarashi, T.; Legrand, A.-M.; Cruchet, P.; Chinain, M.; Fujita, T.; Naoki, H. Structural elucidation of ciguatoxin congeners by fast-atom bombardment tandem mass spectroscopy [5]. J. Am. Chem. Soc. 2000122, 4988–4989.

3) The authors use the nomenclature of ciguatoxins in Pacific regions, like P-CTX1, P-CTX2, and P-CTX3. I recommend using the nomenclature suggested by Prof. Yasumoto, who has first reported the chemical structure of ciguatoxin.   Instead of P-CTX, use CTX1; instead of P-CTX2, use 52-epi-54-deoxyCTX1B; instead of P-CTX3, use 54-deoxyCTX1B. There are two reasons that I recommend to use Yasumoto’s nomenclature.  One is Yasumoto discovered the first structure of CTX, we should follow his recommendations.   Another reason is although CTX1B was first discovered in the Pacific but also later discovered in the Atlantic.  So, the name Pacific CTX is not appropriate.   This is my personal opinion but please consider it.

4) The author mentioned CigusCheck discovered by Hokama et al. However the results of the method have been in a controversy (for example, please see Toxicon 46 (2005) 563-571.)  I recommend the authors include recently reported sandwich ELISA detection of CTXs by Tsumuraya and Hirama. (for example, Analytical Chemistry 90 (2018), 7318)

Reviewer 2 Report

The Manuscript entitled “Epidemiology and toxicology of Ciguatera poisoning in the Colombian Caribbean” aims to give an overview on the Ciguatera Fish Poisonig (CFP) phenomenon in Colombia. CFP is foodborne illness due to consumption of large reef fish that can bioaccumulate certain toxins that in this way reach humans.  In principle, the topic of the manuscript is very interesting and even useful considering that CFP, once confined to tropical regions, is now  spreading also to temperate areas, and so enlarging information available on this emerging issue is crucial. Anyway,  despite all this, the manuscript at the moment is at the stage of a first draft, needing important  reorganizations: information are given in a confusing way, and even if I’m not a native speaker I strongly suggest improving the language quality!

I used text callout tool for reviewing and in the attached pdf you’ll find a number of specific comments that can be explicative of my opinion.
